# The Diverse Roles of TNNI3K in Cardiac Disease and Potential for Treatment

**DOI:** 10.3390/ijms22126422

**Published:** 2021-06-15

**Authors:** Caroline Pham, Noelia Muñoz-Martín, Elisabeth M. Lodder

**Affiliations:** Department of Clinical and Experimental Cardiology, Heart Center, University of Amsterdam, Amsterdam UMC, 1105 AZ Amsterdam, The Netherlands; c.pham@amsterdamumc.nl (C.P.); noeliamumar@gmail.com (N.M.-M.)

**Keywords:** TNNI3K, supraventricular arrhythmias, cardiomyopathy, cardiac regeneration, conduction disease, hypertrophy, myocardial infarction, phosphorylation

## Abstract

In the two decades since the discovery of TNNI3K it has been implicated in multiple cardiac phenotypes and physiological processes. TNNI3K is an understudied kinase, which is mainly expressed in the heart. Human genetic variants in *TNNI3K* are associated with supraventricular arrhythmias, conduction disease, and cardiomyopathy. Furthermore, studies in mice implicate the gene in cardiac hypertrophy, cardiac regeneration, and recovery after ischemia/reperfusion injury. Several new papers on TNNI3K have been published since the last overview, broadening the clinical perspective of *TNNI3K* variants and our understanding of the underlying molecular biology. We here provide an overview of the role of TNNI3K in cardiomyopathy and arrhythmia covering both a clinical perspective and basic science advancements. In addition, we review the potential of TNNI3K as a target for clinical treatments in different cardiac diseases.

## 1. Introduction

Troponin I interacting kinase (TNNI3K), encoded by *TNNI3K*, was discovered by Zhao et al. in 2003 [1]. The gene is localized in human chromosome 1 (1p31.1) and is mainly expressed in the heart. Minor *TNNI3K* expression was reported in the brain and testis [2]. In the heart, *TNNI3K* demonstrates a cardiomyocyte (CM) preferential expression over ventricular fibroblasts [2]. BLAST analysis showed high similarity to the integrin-linked kinase (ILK), which is involved in cardiac growth, contractility, and repair. The *TNNI3K* gene is highly conserved among species, including mice and humans [1]. However, different inbred mouse strains have different cardiac *Tnni3k* expression levels [3,4]. Several mouse strains carry a natural loss of function allele and yet appear to be healthy. Hence, absence of *Tnni3k* is well tolerated in mice, suggesting that the murine *Tnni3k* gene is not essential for life.

The full-length cDNA of *TNNI3K* consists of 3420 base pairs and encodes a protein of 835 amino acids with a molecular mass of approximately 93 kDa. The protein is composed of four domains, a coiled-coil domain at the N-terminal part and a central kinase domain flanked by an ankyrin repeat domain, and a serine-rich domain at the C-terminal part (Figure 1A). The C-terminal part negatively regulates the kinase activity, while the ankyrin repeat domain is necessary for (auto-)phosphorylation [5]. The ankyrin repeat domain likely plays a role in protein interaction as it has been demonstrated to be a protein binding target [6]. According to in vitro kinase assays, the kinase domain exhibits dual-specific kinase activity with both tyrosine and serine/threonine amino acids [5].

Conjoined transcripts of *TNNI3K* with the neighboring gene *FPGT* have been found in humans, potentially resulting in a fusion protein of 949 amino acids (approximately 105 kDa) [7]. However, this protein is absent from the large dataset of the genotype expression (GTEx) project and no evidence of this read-through gene has been further reported.

## 2. TNNI3K Upstream Regulators and Downstream Targets

Despite the multiple studies on TNNI3K, very little is known about its regulation and direct interactors. In 2008, the murine *Tnni3k* cDNA was cloned and Mef2c binding sites were found in the promoter region [8]. It was shown that mutations in these Mef2c binding sites reduced the transcriptional activity of *Tnni3k*, pointing to this transcription factor as a critical regulator of *Tnni3k* expression in mice [8]. However, more studies are needed to determine Tnni3k upstream regulators.

Several downstream targets for TNNI3K have been proposed, but the majority of these targets remain unknown. The name of the protein is due to its interaction with troponin I (cTnI) encoded by *TNNI3* which was described by Luft et al. in 2003 in a yeast 2-hybrid system and later confirmed by Wang et al. using an in vitro model of HEK293T [9,10]. In 2007, the antioxidant protein 1 (AOP-1) was described as another possible interactor, which binds the ankyrin domain of TNNI3K and downregulates its kinase activity [5]. Other studies suggested P38 and PKA as downstream targets of Tnni3k but it is unknown whether there is a direct interaction in vitro or in vivo [2,11]. It is important to highlight that none of these targets have been validated in animal models and many new possible targets remain to be identified.

## 3. TNNI3K and Cardiac Hypertrophy

In mammals, postnatal heart growth is mainly due to the increased size of individual CMs. During adulthood, certain stress conditions such as cardiac overload can lead to CM growth, which is an essential mechanism to maintain cardiac function [12]. This phenomenon is known as cardiac hypertrophy and can either be physiologic or pathologic. Physiological hypertrophy is characterized by preserved cardiac function and can be reversible, whilst pathological hypertrophy is associated with fibrosis, inflammation, cardiac remodeling, and an increased risk of heart failure [12].

Several studies have associated increased TNNI3K levels with cardiac hypertrophy. An in vitro study demonstrated that TNNI3K mRNA expression levels were increased in hypertrophy induced CMs [13]. In addition, adenovirus-mediated overexpression of human TNNI3K promoted increased cell area and sarcomere organization in cultured CMs [13]. In a transverse aortic constriction (TAC) rat model, the expression of myocardial Tnni3k was significantly increased 2 weeks after TAC. Therefore, this suggests that this kinase might be involved in the development of murine cardiac hypertrophy in vivo [10]. Furthermore, human *TNNI3K* overexpression in adult rat ventricular CMs [10] led to a significant increase of cell shortening when compared to control CMs, suggesting that TNNI3K also influences CM contractility. In 2015, a study showed that miR-223 directly targets *TNNI3K* and suppresses cardiac hypertrophy [14]. In vitro experiments, using hypertrophy induced neonatal rat CMs, showed that miR-223 mRNA expression was significantly reduced in hypertrophic CMs. In contrast, miR-223 overexpression led to reduced hypertrophic marker expression levels.

Wheeler et al. generated the first transgenic mouse carrying the human *TNNI3K* under the a-MHC cardiac-specific promoter (TNNI3K^tg^) [3]. The DBA/2J, a mouse strain that naturally expresses low levels of the *Tnni3k* transcript, was used as a genetic background for this model [3,4]. The authors reported accelerated disease progression after TAC in *TNNI3K* transgenic mice, showing greater diastolic and systolic dysfunction and significantly reduced fractional shortening, compared to wild-types (WT) [3]. In a double transgenic heart failure mouse model expressing both Calsequestrin (*Csq*) and *TNNI3K* (TNNI3K^tg^/Csq^tg^), *TNNI3K* was identified as a modifier of disease severity [3]. Studies analyzing TNNI3K^tg^ mice (at 2 months of age) reported increased CM size and fibrosis when compared to DBA/2J mice, suggesting that TNNI3K overexpression induced pathological hypertrophy. Additionally, a mouse model overexpressing a kinase death version of the protein was generated (TNNI3K-KD^tg^) to elucidate the relevance of the kinase activity in the found phenotypes [15]. TNNI3K-KD^tg^ mice presented signs of cardiac hypertrophy although it was not as high as in the TNNI3K^tg^ model. When TAC was performed in those models an exacerbation of disease progression (greater fibrosis and hypertrophy) was observed in TNNI3K^tg^ but not in TNNI3K-KD^tg^. These results demonstrated that the kinase activity is of great importance to drive the hypertrophic phenotype [15]. Two more transgenic models have been created in different genetic backgrounds. A cardiac-specific TNNI3K overexpression in C57Bl/6J [10] and a TNNI3K overexpression model in FVB/N [2] (Table 1). Under resting conditions, these models did not show overt cardiomyopathy. Aged TNNI3K^tg^ C57BL/6J mice showed mild concentric remodeling demonstrated by an increase in heart weight to body weight ratio without cardiac dysfunction [10]. The differences found across these three human TNNI3K transgenic mouse models might be due to several factors. First, the absolute level of transgene expression achieved in each model is different, which might be the main explanation for the different phenotypes. Second, the different genetic backgrounds presented different basal expression levels of murine Tnni3k, and most likely of its (unknown) downstream targets as well, which may also lead to disparate responses.

A congenic mouse line was generated (DBA.AKR-*Hrtfm2*) to replicate physiological levels of Tnni3k expression in a DBA background [3]. This mouse model carries the Hrtfm2 allele (20Mb) of the AKR background, containing the *Tnni3k* gene, and DBA alleles throughout the rest of the genome. No phenotype was reported on DBA.AKR-Hrtfm2 [3]. Nevertheless, a more detailed study would be of great interest to elucidate the relation of normal expression levels of Tnni3k with the development of cardiac hypertrophy.

A complete knockout (KO) model of Tnni3k (Tnni3k-KO) was generated in a C57BL/6J background. Initially, no phenotype was reported regarding cardiac structure and function [16]. However, in a recent study, the authors described a subtle concentric remodeling that appeared at 2 months of age but without affecting the heart weight to body weight ratio [11]. Nevertheless, mice lifespan was normal suggesting that the loss of function of Tnni3k was not leading to pathogenic remodeling in C57BL/6J inbred mouse strains [11].

Overall, Tnni3k seems to be orchestrating part of the hypertrophic response under cardiac overload conditions in mice. Therefore, overexpression of TNNI3K in murine models leads to a stronger hypertrophic phenotype in vitro and in vivo. Nonetheless, it is still unclear whether TNNI3K is capable of inducing hypertrophy in non-stressing conditions. Due to the contradictory results in the overexpression transgenic models, further analyses in the congenic mouse model are needed to clarify the actual function of the protein.

## 4. TNNI3K in Myocardial Infarction and Heart Failure

Ischemic cardiomyopathy (ICM) is a condition describing left ventricular (LV) dysfunction due to cardiac damage as a result of myocardial ischemia (MI) and is considered the most common cause of heart failure [18]. Patients suffering from ICM have shown increased TNNI3K levels in their LV cardiac tissue, suggesting that there is an association between ICM and TNNI3K expression [2]. However, still little is known about the mechanism of TNNI3K in ICM.

Modulation of TNNI3K activity as a potential treatment for ICM has been investigated in C57BL6/J mice that were subjected to MI by left anterior descending artery (LAD) ligation. Subsequently, injection with TNNI3K-overexpressing pluripotent P19CL6 cells into the infarcted hearts demonstrated improved LV performance (by measuring ejection fraction (EF) and LV end-diastolic and end-systolic dimensions) and smaller infarct size compared to injections with vehicle-medium-only and FLAG-only cells [19]. The authors indicated that TNNI3K-dependent cell therapy had beneficial effects on murine cardiac function after MI.

Besides persistent MI, the role of TNNI3K was investigated in ischemia/reperfusion (I/R) injury via coronary artery occlusion/release in TNNI3K^tg^ DBA/2J mice [2]. TNNI3K^tg^ mice subjected to I/R injury demonstrated larger cardiac infarction and elevated cTnI plasma than TNNI3K-KD^tg^ mice and WT littermates. This indicates that enhanced TNNI3K kinase activity is associated with greater infarct size and higher cTnI plasma levels in I/R injury [2,20]. Following studies in Tnni3k-KO undergoing I/R supported this premise by demonstrating significantly smaller cardiac infarcts and lower levels of cTnI, cardiac troponin T, and myosin light chain compared to WT C57BL/6J littermates. In contrast, Tnni3k-KO mice presented comparable LV dysfunction, hypertrophy, and remodeling as in a permanent occlusion (non-reperfused) model of MI. This implies detrimental cardiac effects of TNNI3K in response to reperfusion injury [2].

An important element of I/R injury is the accumulation of oxidative stress by elevated reactive oxygen species (ROS) (e.g., superoxide and peroxide), leading to cell death [21]. Post I/R injury ROS production was assessed by measuring superoxide in TNNI3K^tg^ and TNNI3K-KD^tg^ mice. Increased superoxide production was observed in TNNI3K^tg^ mouse hearts compared to hearts of WT animals, which was derived from mitochondrial dysfunction. On the other hand, hearts from TNNI3K-KD^tg^ mice did not show elevated superoxide levels. These results suggest that high TNNI3K expression levels promote ROS generation, which contributes to a worse cardiac phenotype after cardiac I/R injury [2]. The association between TNNI3K and ROS was further supported by a study by Gan et al. suggesting a co-acting process between Tnni3k and ROS in the context of cellular ploidy, which will be discussed in the following section [11]. All in all, in response to I/R injury, mice overexpressing TNNI3K demonstrated a worsened cardiac phenotype and mitochondrial dysfunction compared to TNNI3K-KD^tg^, Tnni3k-KO, and non-transgenic mice.

## 5. TNNI3K in Cardiac Regeneration

In 2011, the transient regenerative capacity of the neonatal mouse heart was first described [22]. CM cell cycle arrest and loss of regeneration ability occur after postnatal day 7 [22]. However, neonatal heart size is duplicated in a few days due to hypertrophy [23]. An increase in CM ploidy results in binucleation of CMs in mice and nuclear polyploidization in humans, which causes increased cell size explaining the growth in postnatal hearts [24]. WT zebrafish hearts, which fully regenerate after myocardial injury, contain 1% polyploid CMs. Induced CM polyploidy in this model leads to the loss of myocardial regeneration after injury [25]. Therefore, understanding how ploidy is determined in the heart could open a new path to treat myocardial injury in humans.

In this regard, Fuller et al. in 2015 described significantly higher TNNI3K expression levels in adult versus neonatal rat CMs, suggesting that the kinase might be relevant for postnatal heart maturation [26]. In vitro studies suggested a role of TNNI3K during embryonic CM differentiation, but further studies are needed to confirm those findings [27].

In 2017, a study on 120 inbred mouse strains described a natural variation in the frequency of mononuclear CMs (MNCMs) (ranging from 2.3 to 15%). Genome-wide association analysis identified Tnni3k as one of the genes driving that variation [16]. Higher levels of Tnni3k were associated with lower percentages of diploid MNCMs and worse EF after MI [11,17]. Conversely, the mouse strains that recovered almost a normal EF after MI had the lowest levels of Tnni3k and up to 15% of diploid MNCMs. In addition, natural variation in Tnni3k expression across strains, as measured by microarray, was negatively correlated with MNCM frequency. Tnni3k-KO mice in a C57Bl/6 background (low frequency of MNCMs) did induce a 2.5-fold increase in MNCM percentage in resting conditions, reaching up to 5%. However, this did not improve EF or reduce scar size after MI compared to WT C57Bl/6J mice [16]. The authors suggested that the observed increase of up to 5% of MNCMs within Tnni3k-KO hearts was insufficient to ameliorate cardiac recovery after MI.

The number of MNCMs in mice was found to be dependent on the kinase activity of Tnni3k [11,17]. TNNI3K-KD^tg^ (C57Bl/6J genetic background) and Tnni3k-KO mice showed a similar rise in mononuclear CM percentage when compared to *Tnni3k* heterozygous mice. The mechanism that influences the ploidy remains unclear. The authors proposed that Tnni3k converges with oxidative stress to induce the polyploidization of CMs during postnatal cardiac maturation [11]. However, there was no evidence on how Tnni3k and oxidative stress interact. Further studies focusing on the targets of Tnni3k would be of relevant interest in this regard.

Altogether, these findings demonstrated the regulation of CM ploidy by the kinase activity of Tnni3k in mice, which is directly related to the regeneration capacity of the heart. The molecular mechanisms orchestrating this relation remain unknown and more studies focusing on the direct targets of Tnni3k could open a light in this sense. It would be of great interest to know if the natural variation in expression levels of Tnni3k occurs in humans and whether it is also influencing the ploidy and capacity to recover after an MI.

## 6. TNNI3K in Cardiac Conduction

Electrical conduction between atria and ventricles can be visualized as the PR interval on the surface electrocardiogram (ECG). Insufficient propagation of the electrical signal between atria and ventricles can result in an atrioventricular (AV) delay, characterized by a prolonged PR interval. PR interval prolongation is a predictor of atrial fibrillation (AF) and other atrial and ventricular arrhythmic disorders associated with sudden cardiac death (SCD) [28,29].

The PR interval was detected as a quantitative trait locus (QTL) on mouse chromosome 3. By eQTL mapping, *Tnni3k* was identified as a promising candidate gene influencing PR interval variances. To validate *Tnni3k*-dependent PR interval regulation in vivo, different mouse strains were examined. Different WT inbred mouse strains express different *Tnni3k* levels [4]. DBA/2J mouse hearts naturally express the *Tnni3k* transcript at very low levels, whereas other mouse lines such as AKR/J and C57BL/6 show higher cardiac *Tnni3k* expression [3,4]. Mice expressing low *Tnni3k* levels have relatively shorter PR intervals compared to mice with higher *Tnni3k* expression. In addition, both DBA/2J mice overexpressing hTNNI3K and congenic (DBA.AKR.hrtfm2) mice, which harbor the AKR/J ‘high-*Tnni3k* expression’ haplotype in the DBA/2J genetic background, demonstrated a prolonged PR interval compared to WT DBA/2J mice. Furthermore, TNNI3K^tg^ mice showed a significant increase in QRS duration and a decreased heart rate compared to WT DBA/2J mice. Western blot analysis demonstrated Tnni3k protein expression in both atria and ventricles of AKR/J mice, where it was found to be higher expressed in the atria than in the ventricles. As expected, TNNI3K protein expression was undetectable in DBA/2J hearts [4]. *Tnni3k* expression levels correlate to PR interval durations in mice, indicating a role for Tnni3k in AV conduction [4]. Nevertheless, the mechanism in which Tnni3k affects AV conduction is not yet clarified. Considering multiple mechanisms are involved in cardiac conduction, we suggest that Tnni3k could affect gap-junctional coupling or CM activation leading to altered AV conduction.

In summary, different inbred mouse strains show different *Tnni3k* expression patterns, from which high Tnni3k expression in mice is correlated to worse cardiac function, conduction, and regeneration compared to low Tnni3k expression (Table 1; Figure 2).

## 7. TNNI3K in Human Genetics

In humans, several variants in the *TNNI3K* gene have been associated with altered kinase activity and various cardiac phenotypes, including conduction disease (CCD), dilated cardiomyopathy (DCM), supraventricular arrhythmia, and SCD (Figure 1B) [30,31,32,33,34]. The first paper that associated *TNNI3K* with cardiac disease describes a multi-generation family carrying a missense variant in *TNNI3K* (c.1577G>A) co-segregating with atrial tachyarrhythmia, CCD, and DCM. All seven family members affected by the cardiac phenotype were heterozygous carriers of this variation, whereas the unaffected family members were not. This variant resulted in the TNNI3K-p.Gly526Asp substitution in the kinase domain, leading to the hypothesis that this variant would hinder TNNI3K kinase activity. In silico structural modeling of this variant showed that this variant did not change the monomeric TNNI3K structure, but did enhance the binding affinity [30]. A TNNI3K auto-phosphorylation assay revealed abolished TNNI3K auto-phosphorylation in cells harboring the TNNI3K-p.Gly526Asp variant, suggesting reduced TNNI3K kinase activity [32]. When evaluating the ventricular myocardium of the affected individuals, reduced TNNI3K expression levels were observed. In addition, enlarged CMs and mild patchy interstitial fibrosis were identified. Ventricular tissue examined by transmission electron microscopy also revealed myofilament loss, mitochondriosis, and intranuclear inclusions [30].

A second missense variant in *TNNI3K* was identified in a family presenting with CCD, congenital junctional ectopic tachycardia, and atrioventricular nodal re-entrant tachycardia (AVNRT) [31]. Via whole exome sequencing, the TNNI3K-c.1615A>G (TNNI3K-p.Thr539Ala) variant was identified. Affected individuals were heterozygous carriers of the variant, which was inherited from the affected mother [31]. Like TNNI3K-p.Gly526Asp, this variant is located in the kinase domain of TNNI3K and showed reduced kinase activity in HEK293 cells transfected with the TNNI3K-p.Thr539Ala variant compared to WT TNNI3K [32].

More recently the TNNI3K variant (TNNI3K-c.2302G>A, p.Glu768Lys) was reported in three independent three-generation families. This missense variant was identified as the causal variant for supraventricular arrhythmias, AVNRT, CCD, DCM, and SCD. Unlike the two missense variants described above TNNI3K (p.Gly526Asp and p.Thr539Ala), the TNNI3K-p.Glu768Lys variant is localized at the serine-rich domain. However, despite that this variant does not directly affect the kinase domain, it does alter the kinase function by enhancing TNNI3K auto-phosphorylation levels [32].

Furthermore, two recent papers describe apparent loss of function variants: (i) a splice site variant (TNNI3K-c.333+2T>C), predicted to lead to nonsense-mediated decay (NMD) [33] and (ii) a nonsense mutation in TNNI3K (TNNI3K-c.1441C>T, p.R481X) [34]. Both families had individuals with CCD with or without DCM. Unfortunately, as a result of the limited size of the pedigrees, the genetic evidence linking these variants to the phenotypes is thus far limited. These variants should be considered as variants of unknown significance (VUS), especially considering the number of nonsense variants present in the general population [35].

The most common nonsynonymous TNNI3K variant found in the general population is the c.2123T>C (p.Ile686Thr) missense variant. Based on the gnomAD database, this variant is present at an allele frequency of approximately 1% worldwide and 4% in South and East Asian populations [35]. However, this isoleucine to threonine substitution has not been associated with any cardiac phenotype thus far. The resulting amino acid substitution p.Ile686Thr is hypomorphic for the TNNI3K kinase activity, showing reduced auto-phosphorylation activity to 38% compared to WT TNNI3K protein [17]. Mice harboring the human TNNI3K-p.Ile686Thr variant (I685T/I685T) showed affected myocardial dimensions and elevated numbers of MNCMs in comparison to WT mice. The CM area in I685T/I685T was not changed compared to CMs derived from WT mice, but showed significantly shorter and wider cell dimensions [17]. As variability in human cardiac geometry is known in patients with comparable systolic blood pressure [36], the authors suggested that TNNI3K-p.Ile686Thr and other hypomorphic or null alleles might therefore contribute to this variation in cardiac geometry [17].

In summary, thus far, three missense variants in *TNNI3K* show significant genetic association with CCD, DCM, supraventricular arrhythmia, and SCD in humans (Figure 1B) [30,31,32,33,34]. All three missense variants result in altered TNNI3K auto-phosphorylation, indicating a change in protein function [32]. Nonetheless, the molecular mechanism linking the genotypes and observed phenotypes remain unclear, especially considering the mixed direction of effect on auto-phosphorylation in vitro.

## 8. TNNI3K as a Therapeutic Target for Cardiac Diseases

As described above, high levels of TNNI3K are associated with slower conduction, faster progression of cardiac disease, and worse recovery after reperfusion injury, whereas reduced TNNI3K levels showed beneficial cardiac effects [2,3,4,11,16]. In humans, enhanced TNNI3K kinase activity was found in patients harboring the TNNI3K-p.Glu768Lys variant resulting in CCD, DCM, and AVNRT [32]. Furthermore, no evolutionary restriction on the presence of homozygous loss of function alleles appears to be present in the general population [35]. These findings suggest that a treatment strategy addressing cardiac disease by targeting TNNI3K could potentially be beneficial. To achieve such a strategy, the development of potent selective TNNI3K inhibitory drugs is essential. Although many compounds have been generated and tested for their inhibitory effects on TNNI3K, no clinical trials have yet been performed. Here, we will discuss the most promising TNNI3K inhibitors to date (Table 2) [2,37,38]. As selectivity of kinase inhibitors can be problematic due to cross-reactivity [39], it is anticipated that such compounds initially will mostly prove their use in pre-clinical studies.

Previous studies showed that TNNI3K-KD^tg^ mice did not show adverse effects compared to TNNI3K^tg^ mice, which suggests that parts of the negative cardiac effects are driven by the kinase domain of TNNI3K. Therefore, the search for TNNI3K inhibitors led to the development of two active site–binding small-molecule TNNI3K blocking drugs: GSK329 and GSK854 [2]. GSK329 and GSK854 are highly potent TNNI3K inhibitors demonstrating relatively low IC_50_ values of 10 nM and <10 nM, respectively. However, GSK854 showed a higher selectivity than GSK329. Whereas GSK329 showed more than 50% inhibition in 11 other kinases at 100 nM, GSK854 only demonstrated inhibition of ZAK/MLTK [2]. Moreover, GSK854 presented >100-fold selectivity for TNNI3K over 96% of the tested kinases. This implies that GSK854 is a more potent inhibitor than GSK329 [38].

Both of these TNNI3K-targeting drugs have also been tested in mice as a possible approach for I/R injury. WT C57BL/6 mice subjected to ischemia and treated with TNNI3K inhibitors GSK329 or GSK854 (2.75 mg/kg via intraperitoneal injection) during reperfusion showed a significantly reduced infarct size compared to vehicle control [2]. TNNI3K inhibition also showed a long-term effect on cardiac function as an additional 6 weeks of unlimited access to a GSK854 containing chow diet after I/R showed improved LV function compared to vehicle-treated animals after 2 weeks. After 4 weeks, smaller LV end-diastolic and -systolic dimensions were observed in animals treated with the TNNI3K inhibitor. This suggests a protective response against ventricular remodeling by TNNI3K inhibition. At 6 weeks after I/R injury, GSK854 treated animals showed reduced atrial natriuretic peptide precursor (pro-ANP) levels, smaller CM areas, and less cardiac fibrosis versus control vehicle-treated mice, indicating decreased heart failure progression [2].

In 2015 and 2016, Lawhorn et al. introduced several novel TNNI3K blocking drugs with a 7-deazapurine scaffold [37,40]. Unfortunately, most derivatives lacked the specificity to inhibit the function of only TNNI3K. These compounds were found to also inhibit B-Raf kinase, which is structurally comparable to TNNI3K and has also been linked to several cardiac diseases. However, GSK114 is potentially specific for TNNI3K as it demonstrated a 40-fold selectivity for TNNI3K (IC_50_ = 25 nM) over B-Raf kinase (IC_50_ = 1 µM). At a concentration of 1 µM, GSK114 demonstrated an affinity for seven other kinases: ACK1, B-Raf, GAK, MEK5, PDGFRB, STK36, and ZAK [40,41]. Additional testing for other targets (e.g., GPCRs, ion channels, and proteases) showed an IC_50_ > 1 µM, indicating high selectivity for TNNI3K [40].

Additionally, 35 novel TNNI3K inhibitors with a pyrido[4,5]thieno[2,3-d] pyrimidine template based on GSK854 were developed [42]. From those 35 derivatives, compound 6O (IC_50_ = 410 nM) showed to be the most promising inhibitor. Compound 6O reduced apoptosis and pyroptosis resulting from disrupted p38 signaling in H9c2 rat CMs. C57BL6/J mice subjected to MI presented beneficial cardiac effects after treatment with compound 6O (25 mg/kg via intraperitoneal injection), showing reduced LV inflammatory cells, ventricular dilatation, and fibrosis [42]. However, this compound demonstrated an IC_50_ of 410 nM, which is considerably higher than the other inhibitors.

In summary, the three most potent TNNI3K inhibitors to date are GSK854, GSK114, and GSK329. These drugs are useful leads to define the cardiac biology of TNNI3K and are promising as a new approach in targeted cardiac diseases. In vitro and in vivo use of these TNNI3K inhibitors might be beneficial after cardiac stress.

## 9. Other Potential Roles of TNNI3K

Thus far, the molecular mechanisms underlying the diverse roles of TNNI3K are not fully understood as downstream targets have not yet been identified. However, as described in this review the number of studies associating TNNI3K with different diseases is increasing. In addition to these, several as of yet not replicated studies on viral myocarditis, carcinogenesis, and obesity even further expand the spectrum of potential roles of TNNI3K [43,44,45].

Candidate gene analysis revealed fucose-1-phosphate guanylyltransferase (*Fpgt*) and *Tnni3k* as promising targets for Vms1 (viral myocarditis susceptibility 1), which plays a role in the pathogenesis of coxsackievirus B3 infection, the most common cause of myocarditis [43]. Two coding nonsynonymous single nucleotide polymorphisms (SNPs) in *Fpgt* (rs30203847-C) and *Tnni3k* (rs30712233-T) were identified within the *Vms1* locus. Both SNPs were predicted to be non-conservative and detrimental in function. The coding nonsynonymous SNP in *Tnni3k* was suggested to be damaging, however, secondary to *Tnni3k*-rs49812611. This SNP accounts for aberrant splicing (i.e., NMD) of *Tnni3k* transcripts, resulting in the lack of basal *Tnni3k* expression in A/J or CSS3 mice. Hence, the authors suggested that viral myocarditis is regulated by *Tnni3k* or *Fpgt* via affecting myocardial integrity or cellular immune response, respectively [43].

The locus harboring TNNI3K has been implicated in carcinogenesis [44] these associations are potentially related to the neighboring *FPGT* gen, which is expressed in several tissues. Read-through transcription between the *FPGT* and *TNNI3K* genes results in the FPGT-TNNI3K fusion protein, expressing functional properties from both individual proteins [46]. FPGT plays a major role in the salvage fucosylation pathway, which emerges in the biosynthesis of GDP-fucose. Increased fucosylation levels have been associated with carcinogenesis [47]. Fucosylation is also involved in several cell signaling processes in inflammation and obesity [46].

Multiple studies have reported the relevance of SNPs near or in the *TNNI3K* loci (rs1514175, rs1514176, rs12142020, rs274609), creating a variant in the TNNI3K or FPGT-TNNI3K transcript, which are positively associated with body mass index, eating behavior, and (child-)obesity [45,48,49,50,51]. Considering the role of FPGT in fucosylation, these associations are most likely driven by FPGT rather than TNNI3K [46].

## 10. Discussion

We here reviewed the role of TNNI3K in the scope of cardiac diseases in mice and humans. In summary, mouse models expressing higher TNNI3K levels have been associated with worsening of cardiac phenotypes, less recovery after I/R injury, cardiac remodeling, and slower cardiac conduction. These effects are ameliorated in absence of the kinase function of the protein and by inhibition of TNNI3K through inhibitors. In mice, the absence of TNNI3K is well tolerated, although targeted deletion *Tnni3k* leads to mild alterations of cardiac structure. In humans, the situation is more complex. Variants in *TNNI3K* are associated with supraventricular arrhythmias, cardiomyopathy, SCD, and CCD [30,31,32,33,34]. The role of haploinsufficiency of TNNI3K in cardiac disease is thus far unclear due to a lack of large families and the presence of such variants in the general population. Variants with solid genetic evidence of association with disease, thus far, all impact the kinase function of the protein, as measured by auto-phosphorylation. However, the direction of effect of the observed variants is not clear as both loss of auto-phosphorylation and increased auto-phosphorylation have been observed. Multiple mechanisms have been suggested to explain these observations such as dominant-negative effects or sequestration of the targets of TNNI3K. Future work, delineating the targets of TNNI3K and the molecular and electrophysiological pathways involved downstream of this kinase, is necessary to resolve these controversies. Considering the potential for positive impact on the treatment of cardiac disease through inhibition of this kinase it is essential to establish this in human patients.

## Figures and Tables

**Figure 1 ijms-22-06422-f001:**
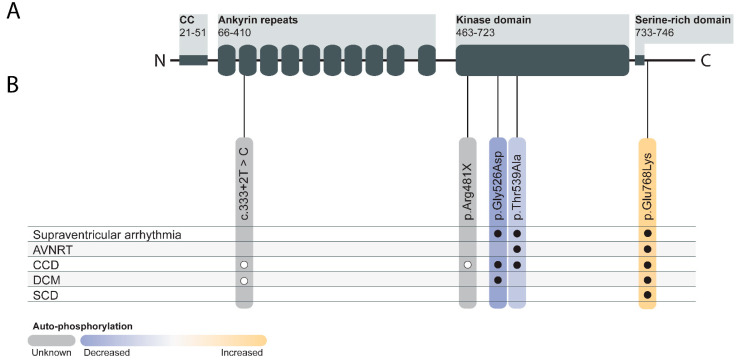
Schematic TNNI3K protein structure and the distribution of genetic variants. (**A**) Schematic TNNI3K protein from the N-terminus to the C-terminus containing a coiled-coil (CC) domain, ankyrin repeats, a kinase domain, and a serine-rich domain. The numbers represent the amino acid location. (**B**) Distribution of TNNI3K variants associated with cardiac diseases and changes in auto-phosphorylation. The color indicates the level of auto-phosphorylation (see color legend). Open circles: phenotypes described with variants of unknown significance; closed circles: phenotypes in families with significant genetic evidence for association with disease. AVNRT: atrioventricular nodal re-entrant tachycardia; CCD: cardiac conduction disease; DCM: dilated cardiomyopathy; SCD: sudden cardiac death.

**Figure 2 ijms-22-06422-f002:**
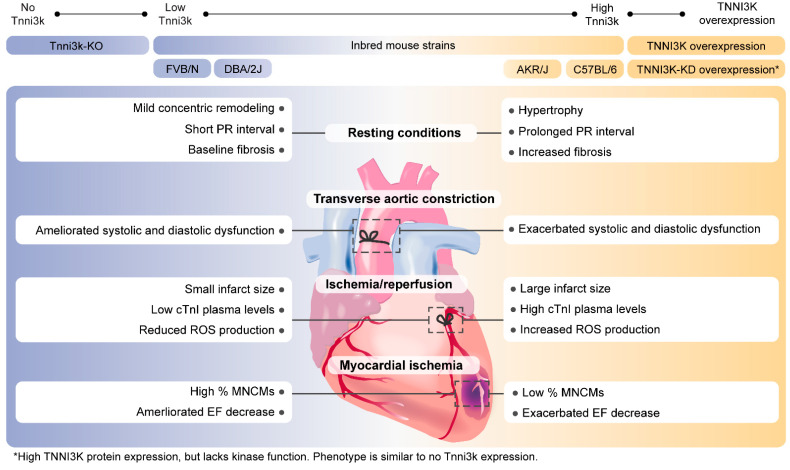
Summary of cardiac phenotypes in mouse models generated to study the function of TNNI3K. The upper panel shows the Tnni3k/TNNI3K overexpression mouse models and inbred mouse strains with their relative Tnni3k expressions. The lower panel summarizes cardiac features in mice expressing relatively low (on the left) and high (on the right) Tnni3k levels in resting conditions, after TAC, I/R injury, or MI. cTnI: Troponin-I; EF: Ejection fraction, MNCMs: Mononuclear cardiomyocytes.

**Table 1 ijms-22-06422-t001:** Overview of discussed mouse models. I/R: Ischemia/reperfusion; MI: Myocardial infarction; TAC: Transverse aortic constriction.

Model Name	Genetic Model	Genetic Background	Intervention	Results	Reference
Congenic	Physiological Tnni3k expression (DBA.AKR.hrtfm2)	DBA/2J	N/A	PR interval prolongation compared to DBA/2J WT	[4]
TNNI3K^tg^	Transgenic human TNNI3K overexpression	DBA/2J	TAC	Greater cardiac dysfunction	[3,15]
			N/A	PR interval prolongation	[4]
		C57BL/6J	N/A	Concentric hypertrophy	[10]
		FVB/N	I/R	Increased infarct size, mitochondrial dysfunction, increased cell death, and ROS production	[2]
TNNI3K-KD^tg^	Transgenic human TNNI3K-p.K489R (kinase-dead) overexpression	DBA/2J	TAC	Similar cardiac dysfunction compared to WT	[15]
		FVB/N	I/R	Smaller infarct size, reduced cTnI levels	[2]
		C57BL/6J	N/A	Increased MNCMs %, shorter and wider CM dimensions	[11,17]
TNNI3K^tg^/Csq^tg^	Transgenic human TNNI3K overexpression with transgenic overexpression of Csq	DBA/2J	N/A	Reduced survival rate, cardiac function, and cardiac dilatation	[3]
Tnni3k-KO	Complete Tnni3k knockout	C57BL/6J	N/A	Mild concentric remodeling, increased MNCMs %, shorter and wider CM dimensions	[11,17]
			MI	Increased CM proliferation in the border zone	[16]
			I/R	Smaller infarct size, reduced cTnI levels, and less cell death	[2]
I685T/I685T	Knock-in allele resulting in Tnni3k-p.Ile685Thr	C57BL/6J	N/A	Increased MNCMs %, shorter and wider CM dimensions	[17]

**Table 2 ijms-22-06422-t002:** TNNI3K inhibitors.

Compound	TNNI3K IC_50_ (nM)	Other Target(s)	Chemical Structure	Reference
GSK854	<10	ZAK/MLTK (tested at 100 nM)	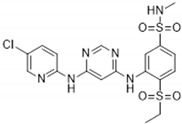	[2]
GSK329	10	Axl, DDR2, Flt1, Flt3, Flt4, KDR, Mer, MuSK, PTK5, TAO2, TAO3 (tested at 100 nM)	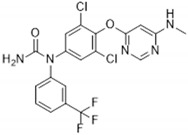	[2]
GSK114	25	ACK1, B-Raf, GAK, MEK5, PDGFRB, STK36, ZAK (tested at 1 µM)	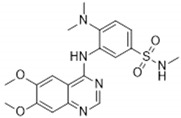	[40,41]
6O	410	TAK1 (tested at 10 µM)	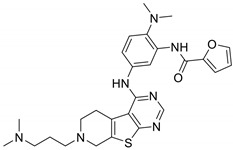	[42]

## Data Availability

Not applicable.

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
