# Peer review of "The Diverse Roles of TNNI3K in Cardiac Disease and Potential for Treatment"

_ijms, 2021, doi:10.3390/ijms22126422_

Round 1
Reviewer 1 Report
This manuscript is a fantastic, very well-written review. The illustrations and charts are highly informative. It focuses on TNNI3K, which is gaining significant attention in the last years in the context of cardiac regeneration and polyploidization.
I have only a few comments.
Line 28 - "high proximity" would indicate that the two genes are close by. If the two proteins are highly similar or are highly conserved, perhaps "proximity" might be a bit confusing.
Line 202. Tnni3k was identified as ONE of the genes associated with the change in number of binucleated cardiomyocytes. The way it is written here suggests it is the only gene associated.
Author Response
Reviewer 1
- Line 28 - "high proximity" would indicate that the two genes are close by. If the two proteins are highly similar or are highly conserved, perhaps "proximity" might be a bit confusing.
We changed “proximity” into “similarity”.
- Line 202. Tnni3k was identified as ONE of the genes associated with the change in number of binucleated cardiomyocytes. The way it is written here suggests it is the only gene associated.
We changed “Genome-wide association analysis identified Tnni3k as the gene driving that variation” into “Genome-wide association analysis identified Tnni3k as one of the genes driving that variation” (Line 222).
Reviewer 2 Report
This manuscript provides a comprehensive review on TNNI3K in cardiac disease. It seems very extensive, and the well-designed tables and figures are helpful for understanding. There are minor suggestions below.
- It would be great to add one more section summarizing the roles of TNNI3K during heart development.
- “TNNI3K as a target for cardiac diseases” -> “therapeutic target” would be clearer.
- The content is understandable, but English language editing and corrections are required throughout the manuscript. Examples: “diverse role” -> diverse roles, “loss-off-function” -> loss of function, “hypertrophic induced CMs” -> hypertrophy induced CMs
Author Response
Reviewer 2
- It would be great to add one more section summarizing the roles of TNNI3K during heart development.
Up to now, no data on TNNI3K in heart development has been published. Therefore, we could not add this section to the manuscript.
- “TNNI3K as a target for cardiac diseases” -> “therapeutic target” would be clearer.
We changed the heading of section 8 “TNNI3K as a target for cardiac diseases” into “TNNI3K as a therapeutic target for cardiac diseases”.
- The content is understandable, but English language editing and corrections are required throughout the manuscript. Examples: “diverse role” -> diverse roles, “loss-off-function” -> loss of function, “hypertrophic induced CMs” -> hypertrophy induced CMs
We revised the manuscript and corrected for inappropriate English language. We corrected errors indicated by the reviewer.